# Interactions between a subset of substrate side chains and AAA+ motor pore loops determine grip during protein unfolding

**Tristan A Bell[1]\*, Tania A Baker[1,2], Robert T Sauer[1]\***

[1]Department of Biology, Massachusetts Institute of Technology, Cambridge, United States; [2]Howard Hughes Medical Institute, Massachusetts Institute of Technology, Cambridge, United States

**Abstract** Most AAA+ remodeling motors denature proteins by pulling on the peptide termini of folded substrates, but it is not well-understood how motors produce grip when resisting a folded domain. Here, at single amino-acid resolution, we identify the determinants of grip by measuring how substrate tail sequences alter the unfolding activity of the unfoldase-protease ClpXP. The seven amino acids abutting a stable substrate domain are key, with residues 2–6 forming a core that contributes most significantly to grip. ClpX grips large hydrophobic and aromatic side chains strongly and small, polar, or charged side chains weakly. Multiple side chains interact with pore loops synergistically to strengthen grip. In combination with recent structures, our results support a mechanism in which unfolding grip is primarily mediated by non-specific van der Waal's interactions between core side chains of the substrate tail and a subset of YVG loops at the top of the ClpX axial pore.

DOI: https://doi.org/10.7554/eLife.46808.001

**\*For correspondence:**
tribell@mit.edu (TAB);
bobsauer@MIT.EDU (RTS)

**Competing interests:** The authors declare that no competing interests exist.

## Introduction

Cells maintain homeostasis by balancing protein synthesis and degradation with growth. When nutrients are available, new proteins are constantly synthesized, whereas damaged, misfolded, or unneeded proteins are degraded. Regulated degradation typically requires a protein-unfolding motor of the AAA+ family (ATPases associated with various cellular activities) that associates with a self-compartmentalized protease (e.g., ClpX with ClpP, the 19S regulatory particle with the 20S proteasome) or is genetically tethered to a protease (e.g., Lon, FtsH, Yme1) (*Sauer and Baker, 2011*; *Olivares et al., 2016*; *Glynn, 2017*). When challenged with degrading a folded substrate, AAA+ motors couple ATP hydrolysis to mechanical motion that overcomes the resistance of the folded domain. Despite broad consensus on the overall mechanism of protein unfolding, it is largely unknown how interactions between a AAA+ motor and its substrate produce grip, the ability for the motor to maintain hold of the substrate while applying an unfolding force.

ClpX is a ring-shaped AAA+ homohexamer that functions autonomously in protein remodeling in bacteria and eukaryotic organelles and also associates with ClpP tetradecamers to form the ATP-dependent ClpXP protease (*Baker and Sauer, 2012*). Substrates are targeted to ClpX or ClpXP by N- or C-terminal peptide tails (also called degradation tags or degrons), which initially bind in the ClpX axial pore. Proteins marked with a sequence-defined degron or post-translational modification can be recruited to the AAA+ protease directly or with assistance from auxiliary adaptors (*Sauer and Baker, 2011*; *Trentini et al., 2016*). For example, during rescue of stalled ribosomes in *Escherichia coli*, the 11-residue ssrA tag is appended to the C-terminus of abortive protein products

**eLife digest** Proteins are the workhorses of the body, fulfilling many roles essential for life processes. These molecules are made up of hundreds or thousands of small units called amino acids, which attach to each other to form a long chain. The exact sequence of amino acids determines how the protein will then fold to acquire its final, three-dimentional shape.

Enzymes called proteases can degrade unneeded or faulty proteins so that the amino acids can be recycled. For instance, in bacteria, the AAA+ protease ClpXP can recognize and 'grab' specific patterns of amino acids at the ends of a protein. This molecular machine then tugs on the segment and unfold the protein, the way a ball of yarn unwinds when pulled from one end. The unfurled protein is then fed into a different section of ClpXP, where it is chopped into short segments for recycling.

ClpXP is the best-characterized enzyme amongst AAA+ proteases. However, it is still unclear how it can grip target proteins tightly enough to allow unfolding. To investigate, Bell et al. attached different patterns of 12 amino acids to the end of a folded protein. How well ClpXP grasped each of these proteins was then measured in bacteria and in test tubes. This revealed that ClpXP attaches to six to eight amino acids at a time, suggesting that only part of the enzyme clasps on the protein. Large amino acids are better gripped than small amino acids, similar to how a knotted string is easier to hold than a smooth rope. Amino acids that are electrically charged also interfere with ClpXP attaching to the protein. Finally, ClpXP grasps multiple amino acids at the same time, which dramatically increases grip strength.

Many proteins, including some found in viruses, use 'slippery' patterns of amino acids to avoid being gripped and unfolded by proteases. By understanding how different patterns of amino acids are grasped, it may someday be possible to engineer enzymes able to target dangerous proteins.

DOI: https://doi.org/10.7554/eLife.46808.002

(*Keiler et al., 1996*), allowing ClpXP to recognize and degrade the attached protein (*Gottesman et al., 1998*; *Farrell et al., 2005*).

The ssrA tag and other degron tails interact with ClpX loops that line the axial pore. A Tyr-Val-Gly (YVG) sequence in the pore-1 loop is critical for substrate binding, unfolding, and translocation (*Siddiqui et al., 2004*; *Martin et al., 2008a*; *Martin et al., 2008b*; *Iosefson et al., 2015*). Other AAA+ unfolding motors contain related pore-1 loops and mutation of these loops typically abolishes function (*Yamada-Inagawa et al., 2003*; *Schlieker et al., 2004*; *Hinnerwisch et al., 2005*; *Park et al., 2005*). In several AAA+ unfolding motors, the pore-1 loops adopt a spiral staircase conformation within the pore, which facilitates multivalent interaction with the bound peptide tail (*Monroe et al., 2017*; *Gates et al., 2017*; *Puchades et al., 2017*; *de la Peña et al., 2018*; *Majumder et al., 2019*; *Dong et al., 2019*; *White et al., 2018*). ATP-dependent conformational changes are thought to draw the tail of a substrate into the pore until a folded domain too large to transit the pore impedes progress. For ClpXP, repeated cycles of ATP hydrolysis are then required to unfold the substrate and to translocate the polypeptide through the pore and into ClpP for degradation (*Kenniston et al., 2003*; *Aubin-Tam et al., 2011*; *Maillard et al., 2011*; *Sen et al., 2013*; *Cordova et al., 2014*).

How the amino acids in the bound substrate tail contribute to grip during unfolding remains poorly understood. When degrading unfolded substrates, the rate of substrate translocation through ClpXP is largely insensitive to amino-acid charge, size, or peptide-bond spacing (*Barkow et al., 2009*). In contrast, when directly abutting a folded domain, sequences rich in glycine can result in failed unfolding by ClpXP or the 26S proteasome, leading to release of partially processed intermediates (*Lin and Ghosh, 1996*; *Levitskaya et al., 1997*; *Sharipo et al., 2001*; *Hoyt et al., 2006*; *Daskalogianni et al., 2008*; *Too et al., 2013*; *Kraut, 2013*; *Vass and Chien, 2013*). Abortive unfolding caused by Gly-rich motifs occurs as a result of slower domain unfolding rather than rapid substrate dissociation, suggesting that these motifs bind normally but are gripped poorly during unfolding (*Kraut, 2013*; *Kraut et al., 2012*). These results suggest that AAA+ motors struggle to efficiently grip sequences with very small side chains. Alternatively, sequence complexity rather than composition may dictate grip strength (*Tian et al., 2005*).

Here, we use green fluorescent protein (GFP) reporter substrates to interrogate the contributions of individual amino acids in the peptide tail to grip strength by ClpXP. In degradation assays performed in vivo and in vitro, we observe that substrate grip by ClpX is primarily mediated by interactions with a block of five amino acids, located two to six residues from the native GFP domain. Through systematic mutation, we characterize the ability of each amino acid to promote ClpX grip, and find that aromatic and large hydrophobic residues are gripped well, whereas charged and polar residues impair grip. Finally, we analyze synergistic contributions of multiple residues to unfolding, and show that contacts with more than one side chain lead to stronger grip and faster substrate unfolding. Our results provide unprecedented detail into the mechanism by which AAA+ motors grip terminal substrate tails during protein unfolding.

## Results

### Substrate design and degradation assays

To probe grip during substrate unfolding, we used *Aequorea victoria* GFP, as its native structure is highly kinetically stable, unfolding is rate limiting for ClpXP degradation of GFP-ssrA, and the pathway of mechanical unfolding of GFP-ssrA by ClpXP is well characterized (*Maillard et al., 2011*; *Kim et al., 2000*; *Nager et al., 2011*). In our substrates, we truncated GFP at Ile-229, the last amino acid that makes extensive native contacts in multiple crystal structures (*Ormö et al., 1996*; *Yang et al., 1996*), and added a 12-residue cassette of variable sequence followed by a partial ssrA degron to allow recognition by ClpXP (*Figure 1A*). Given the length of the axial pore (~35 Å; *Glynn et al., 2009*), we reasoned that ClpX should only interact with residues within the cassette region during GFP unfolding.

For studies of intracellular degradation, we used an *E. coli* B strain, which lacks the AAA+ Lon protease; deleted the chromosomal copies of *clpP*, *clpX*, and *clpA*, as ClpAP can also degrade ssrA-tagged substrates (*Gottesman et al., 1998*; *Farrell et al., 2005*); and placed genes encoding ClpX$^{\Delta N}$ and ClpP on a plasmid under arabinose-inducible control (*Figure 1B*; *Guzman et al., 1995*). Despite lacking a family-specific N-terminal domain, ClpX$^{\Delta N}$ supports ClpP-degradation of ssrA-tagged substrates as well as wild-type ClpX but does not interact with many other cellular substrates and adaptors (*Singh et al., 2001*; *Wojtyra et al., 2003*; *Flynn et al., 2003*; *Martin et al., 2005*). GFP substrates were cloned under transcriptional control of a constitutive ProD promoter (*Figure 1B*; *Davis et al., 2011*). To control for ClpXP-independent degradation, we used a pBAD plasmid isogenic to the *clpP*/*clpX*$^{\Delta N}$ vector but lacking these genes (*Figure 1B*). To determine the extent of intracellular GFP degradation, we measured GFP fluorescence after arabinose induction and growth for 35 min.

We first used this system to characterize substrates with different high or low complexity sequences in the 12-residue cassette. Substrate expression levels were sensitive to the cassette sequence, possibly because of effects on mRNA stability or translation, varying by as much as 7-fold (*Figure 1C*). To control for differences in expression when measuring degradation of different substrates, we normalized measurements in the strain expressing ClpX$^{\Delta N}$P to a strain lacking it (*Figure 1D*, *Table 1*). In these experiments, low-complexity tails rich in acidic or basic residues promoted GFP degradation at levels comparable to a high-complexity sequence derived from the human titin$^{I27}$ domain or a sequence of interspersed glycines and alanines (called GA). By contrast, a cassette sequence of twelve glycines (called Gly$_{12}$) resulted in poor degradation.

We purified N-terminally His$_6$-tagged variants of these substrates and performed Michaelis-Menten analysis of steady-state ClpX$^{\Delta N}$P degradation in vitro (*Figure 1E*). Degradation with the Gly$_{12}$ cassette was too slow to measure, but the GA and acidic cassettes promoted degradation with V$_{max}$ values similar to the titin sequence (*Figure 1F*, *Table 2*). The basic cassette sequence resulted in an intermediate rate of maximal degradation.

The differences between our results in vivo and in vitro suggest that the endpoint assay in vivo has an upper limit and cannot differentiate rates once the pool of cellular GFP has been degraded. About 20% of the Gly$_{12}$ substrate appeared to be degraded in vivo, whereas no degradation was seen in vitro. Maturation of the GFP chromophore lags protein folding (*Reid and Flynn, 1997*), and thus degradation of immature non-fluorescent GFP would not be detected in our cellular assay. It is possible that solutes or macromolecular crowding in the cell enhance ClpXP activity or make GFP

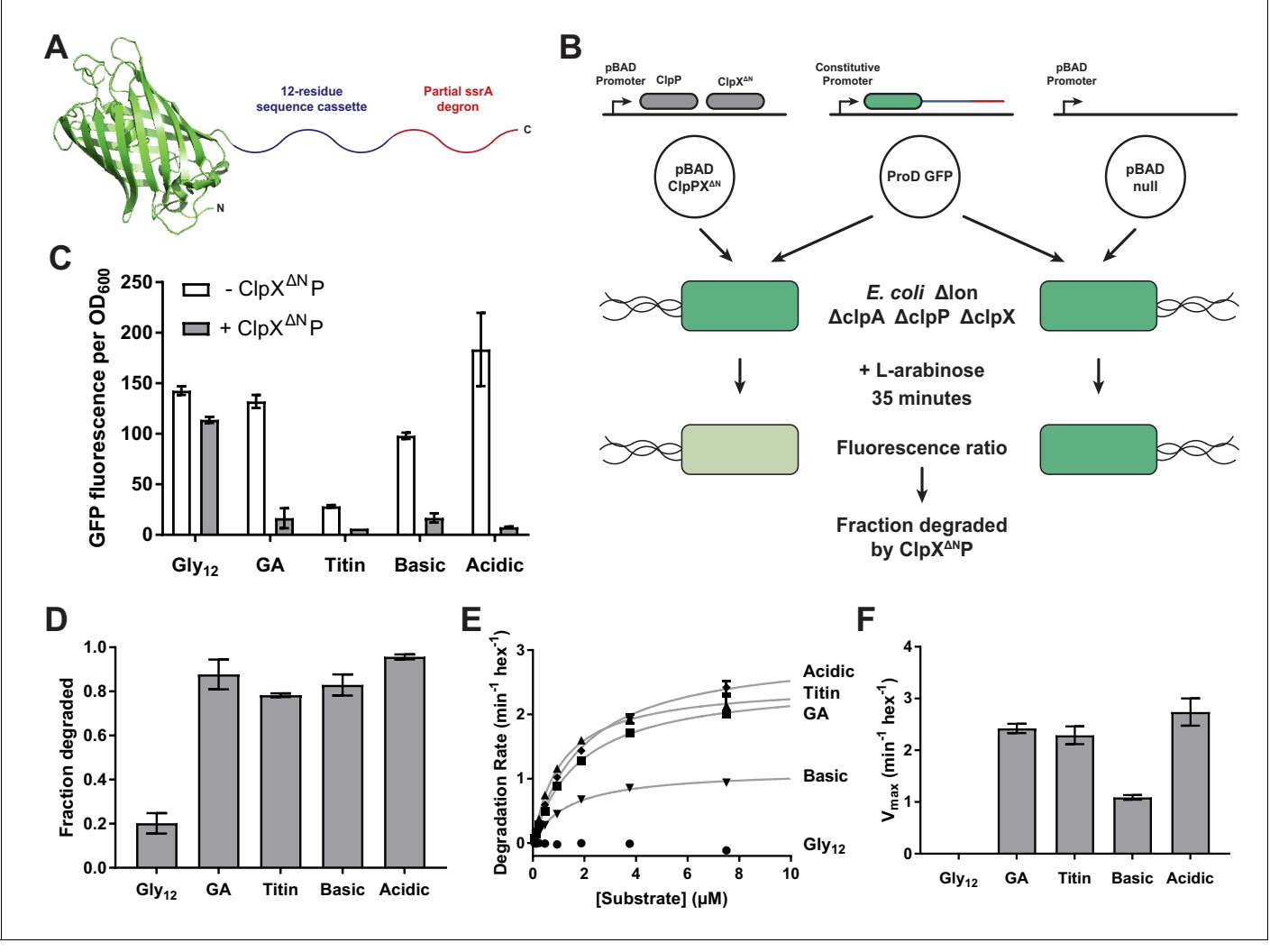

**Figure 1.** Effects of cassette sequence on GFP unfolding and degradation. (A) Starting at the N terminus, substrates contained residues 1–229 of *A. victoria* GFP (PDB 1GFL, *Yang et al., 1996*), a cassette with 12 variable residues, and a partial ssrA degron. (B) Method for measuring intracellular degradation of substrates by ClpX$^{\Delta N}$/ClpP. (C) Cellular fluorescence depends upon ClpX$^{\Delta N}$/ClpP expression and cassette sequence (listed in *Table 1*). (D) Fraction intracellular degradation for substrates bearing different cassettes. (E) Fits of the substrate dependence of degradation in vitro to a hyperbolic Michaelis-Menten equation. (F) $V_{max}$ values for different substrates. In panels, C–F, values represent averages (± S.D.) of three biological replicates.

DOI: https://doi.org/10.7554/eLife.46808.003

easier to unfold. Nevertheless, 12 consecutive glycines inhibit ClpXP unfolding/degradation of GFP both in vivo and in vitro, whereas other low-complexity sequences do not.

## A small stretch of tail residues mediates unfolding grip

We substituted amino acids in the Gly$_{12}$ cassette to identify residues/positions that might improve ClpX grip and thus rates of unfolding and degradation. We first positioned a three-residue Leu-Tyr-Val (LYV) sequence in a sliding window across an otherwise poly-Gly cassette (*Figure 2A*). This tri-peptide sequence was selected because its residues are large and hydrophobic, unlike the surrounding Gly residues. Placing the LYV sequence at positions 2–4, 4–6, or 6–8 (numbered relative to the last residue of the folded domain) improved GFP degradation to levels similar to the GA substrate, whereas this tripeptide at positions 8–10 and 10–12 had no substantial effect relative to the Gly$_{12}$ parent (*Figure 2A*, *Table 1*). These results suggest that ClpX grips side chains within the first eight residues of the substrate tail during unfolding.

**Table 1.** Degradation of variable-tail substrates in the bacterial cytoplasm.

Sequences of all substrate tails tested and the extent of degradation by ClpX$^{\Delta N}$P in *E. coli* after 35 min. For substrates tested in multiple panels, the value presented is from the panel in which they first appear. Values are the average of three biological replicates ± S.D.

| Substrate | Variable tail sequence | Fraction degraded in vivo |
|---|---|---|
| Gly$_{12}$ | GGGG GGGG GGGG | 0.20 ± 0.05 |
| GA | AGAG GGAG AGGA | 0.88 ± 0.07 |
| Titin | HLGL IEVE KPLY | 0.78 ± 0.01 |
| Basic | GKGR GKGR GKGR | 0.83 ± 0.05 |
| Acidic | GEGD GEGD GEGD | 0.96 ± 0.01 |
| LYV$_{2-4}$ | GLYV GGGG GGGG | 0.82 ± 0.03 |
| LYV$_{4-6}$ | GGGL YVGG GGGG | 0.83 ± 0.01 |
| LYV$_{6-8}$ | GGGG GLYV GGGG | 0.65 ± 0.08 |
| LYV$_{8-10}$ | GGGG GGGL YVGG | 0.23 ± 0.01 |
| LYV$_{10-12}$ | GGGG GGGG GLYV | 0.2 ± 0.1 |
| Tyr1 | YGGG GGGG GGGG | 0.3 ± 0.1 |
| Tyr2 | GYGG GGGG GGGG | 0.49 ± 0.08 |
| Tyr3 | GGYG GGGG GGGG | 0.80 ± 0.01 |
| Tyr4 | GGGY GGGG GGGG | 0.80 ± 0.02 |
| Tyr5 | GGGG YGGG GGGG | 0.8 ± 0.1 |
| Tyr6 | GGGG GYGG GGGG | 0.4 ± 0.1 |
| Tyr7 | GGGG GGYG GGGG | 0.32 ± 0.05 |
| Tyr8 | GGGG GGGY GGGG | 0.20 ± 0.03 |
| Ala4 | GGGA GGGG GGGG | 0.28 ± 0.03 |
| Arg4 | GGGR GGGG GGGG | 0.39 ± 0.03 |
| Asn4 | GGGN GGGG GGGG | 0.23 ± 0.01 |
| Asp4 | GGGD GGGG GGGG | 0.20 ± 0.02 |
| Cys4 | GGGC GGGG GGGG | 0.35 ± 0.02 |
| Glu4 | GGGE GGGG GGGG | 0.27 ± 0.02 |
| Gln4 | GGGQ GGGG GGGG | 0.41 ± 0.04 |
| His4 | GGGH GGGG GGGG | 0.26 ± 0.01 |
| Ile4 | GGGI GGGG GGGG | 0.78 ± 0.05 |
| Leu4 | GGGL GGGG GGGG | 0.7 ± 0.1 |
| Lys4 | GGGK GGGG GGGG | 0.36 ± 0.02 |
| Met4 | GGGM GGGG GGGG | 0.6 ± 0.2 |
| Phe4 | GGGF GGGG GGGG | 0.7 ± 0.1 |
| Pro4 | GGGP GGGG GGGG | 0.19 ± 0.03 |
| Ser4 | GGGS GGGG GGGG | 0.24 ± 0.04 |
| Thr4 | GGGT GGGG GGGG | 0.24 ± 0.01 |
| Trp4 | GGGW GGGG GGGG | 0.5 ± 0.1 |
| Val4 | GGGV GGGG GGGG | 0.7 ± 0.1 |
| Ala1 | AGGG GGGG GGGG | 0.41 ± 0.03 |
| Ala1 + 4 | AGGA GGGG GGGG | 0.84 ± 0.03 |
| Ala2 + 4 | GAGA GGGG GGGG | 0.88 ± 0.01 |
| Ala3 + 4 | GGAA GGGG GGGG | 0.88 ± 0.01 |
| Ala4 + 5 | GGGA AGGG GGGG | 0.87 ± 0.02 |

*Table 1 continued on next page*

Table 1 continued

| Substrate | Variable tail sequence | Fraction degraded in vivo |
|---|---|---|
| Ala4 + 6 | GGGA GAGG GGGG | 0.7 ± 0.1 |
| Ala4 + 7 | GGGA GGAG GGGG | 0.51 ± 0.09 |
| Ala4 + 8 | GGGA GGGA GGGG | 0.31 ± 0.04 |
| Ala4 + 9 | GGGA GGGG AGGG | 0.24 ± 0.07 |
| Ala4 + 10 | GGGA GGGG GAGG | 0.29 ± 0.07 |
| Ala4 + 11 | GGGA GGGG GGAG | 0.33 ± 0.06 |
| Ala4 + 12 | GGGA GGGG GGGA | 0.31 ± 0.04 |

DOI: https://doi.org/10.7554/eLife.46808.004

To examine the contributions of individual residues to grip, we constructed another panel of substrates in which a single Tyr residue was placed at each of the first eight tail positions in otherwise all-glycine cassettes (*Figure 2B*). We then tested degradation in vivo. Substrates with a single Tyr at positions 3, 4, and 5 were efficiently degraded, a Tyr at position 2 supported an intermediate level of degradation, and Tyr side chains at other cassette positions supported degradation similar to the $Gly_{12}$ parent (*Figure 2B*, *Table 1*). Thus, four residues appear to contribute the most important grip contacts during unfolding, with tail positions 3–5 being most significant. When we measured degradation of purified substrates in vitro (*Figure 2C*, *Table 2*), single Tyr side chains at positions 2–6 facilitated GFP degradation, with the experimental $V_{max}$ values forming a roughly normal distribution centered around position 4. Again, Tyr side chains at positions 3–5 were most important, Tyr residues at the flanking 2 and 6 positions had small effects, and Tyr side chains at positions 1, 7, or 8 had no discernable effect. Importantly, changing the position of the Tyr side chain altered the maximal rate of unfolding/degradation without substantially affecting $K_M$ for degradation or the ability of substrate to stimulate ATP hydrolysis (*Figure 2D*, *Figure 2—figure supplements 1–2*, *Figure 2—source data 1*). As a result, substrates that were degraded slowly also exhibited a high ATP cost for degradation (*Figure 2E*). In combination, these results support a model in which ClpX preferentially grips the side chains of residues at positions 3–5 during GFP unfolding. Moreover, gripping a single Tyr side chain at one of these positions is sufficient for robust unfolding and degradation of GFP.

## Side-chain grip preferences

Next, we exploited this system to determine how different types of side chains affect ClpX grip. We constructed substrates in which each of the remaining 18 natural amino acids was placed at position 4 of a cassette with glycines at the other 11 positions. These substrates exhibited a wide range of susceptibility to ClpXP degradation in *E. coli* (*Figure 3A*, *Table 1*). In general, tails containing an aromatic or large/branched hydrophobic side chain (Tyr, Phe, Val, Ile, Leu, or Met) promoted the most efficient unfolding and degradation, whereas small and/or polar side chains were least efficient. The inhibitory effects of polarity and charge on grip were most obvious for side chains with similar shapes. For example, Val was one of the best side chains for grip, whereas the isosteric Thr side chain was very poor (*Figure 3B*). Similarly, a polar Gln side chain resulted in better grip than an isosteric but negatively charged Glu side chain (*Figure 3B*).

We also determined steady-state kinetic parameters for degradation of a subset of purified substrates in vitro (*Figure 3C*, *Table 2*). These results largely mirrored results in vivo, with mid-sized or large hydrophobic and aromatic residues promoting the fastest rates of degradation (*Figure 3D*).

Again, Val supported much better degradation than Thr, and Gln promoted significantly faster degradation than Glu in degradation assays in vitro (*Figure 3E*). Further, Ser failed to support GFP degradation while both Ala and Cys facilitated low-level degradation (*Figure 3E*). The Ala-4, Ser-4, Cys-4, Thr-4, Val-4, Glu-4, and Gln-4 substrates at concentrations of 15 µM stimulated the rate of ClpX ATP hydrolysis ~3–4 fold compared to the absence of substrate (*Figure 2—figure supplement 2*, *Figure 2—source data 1*). Thus, each substrate binds ClpX well at this concentration, supporting a model in which the large differences in maximal degradation arise from poor grip caused, at least in part, by differences in side-chain charge and polarity. The maximal degradation rates for these

**Table 2.** Degradation of purified variable-tail substrates in vitro.

Fitted parameters from Michaelis-Menten analysis of substrate degradation by ClpX$^{\Delta N}$P. *No fit* – substrate degradation too slow to be accurately fit. Values are the average of three biological replicates ± S.D.

| Substrate | $V_{max}$ (min$^{-1}$ hex$^{-1}$) | $K_M$ (μM) |
|---|---|---|
| Gly$_{12}$ | *No fit* | |
| GA | 2.4 ± 0.1 | 1.7 ± 0.1 |
| Titin | 2.3 ± 0.2 | 1.1 ± 0.1 |
| Basic | 1.1 ± 0.1 | 1.4 ± 0.1 |
| Acidic | 2.7 ± 0.3 | 1.9 ± 0.2 |
| Tyr1 | *No fit* | |
| Tyr2 | 0.10 ± 0.03 | 0.7 ± 0.2 |
| Tyr3 | 0.7 ± 0.2 | 1.3 ± 0.2 |
| Tyr4 | 1.7 ± 0.1 | 1.9 ± 0.2 |
| Tyr5 | 1.1 ± 0.1 | 1.2 ± 0.1 |
| Tyr6 | 0.08 ± 0.02 | 0.7 ± 0.3 |
| Tyr7 | *No fit* | |
| Tyr8 | *No fit* | |
| Ala4 | 0.13 ± 0.06 | 1.6 ± 0.8 |
| Arg4 | 0.4 ± 0.1 | 1.8 ± 0.4 |
| Asn4 | *No fit* | |
| Asp4 | *No fit* | |
| Cys4 | 0.16 ± 0.04 | 0.9 ± 0.4 |
| Glu4 | 0.11 ± 0.06 | 1.1 ± 0.6 |
| Gln4 | 0.40 ± 0.07 | 1.8 ± 0.3 |
| Ile4 | 1.4 ± 0.3 | 2.2 ± 0.3 |
| Leu4 | 1.3 ± 0.1 | 2.0 ± 0.2 |
| Lys4 | 0.3 ± 0.1 | 2.0 ± 0.7 |
| Met4 | 1.3 ± 0.1 | 2.1 ± 0.3 |
| Phe4 | 1.4 ± 0.2 | 2.5 ± 0.2 |
| Pro4 | *No fit* | |
| Ser4 | *No fit* | |
| Thr4 | 0.10 ± 0.02 | 2.0 ± 0.6 |
| Trp4 | 0.48 ± 0.03 | 1.4 ± 0.1 |
| Val4 | 1.7 ± 0.2 | 2.2 ± 0.2 |
| Ala1 | 0.19 ± 0.04 | 1.8 ± 0.8 |
| Ala1 + 4 | 2.3 ± 0.2 | 2.2 ± 0.1 |
| Ala3 + 4 | 2.4 ± 0.1 | 2.1 ± 0.1 |
| Ala4 + 5 | 1.7 ± 0.2 | 1.5 ± 0.1 |
| Ala4 + 7 | 0.38 ± 0.09 | 0.8 ± 0.2 |
| Ala4 + 9 | 0.09 ± 0.04 | 0.5 ± 0.3 |
| Tyr1 + 4 | 1.5 ± 0.1 | 1.2 ± 0.1 |
| Tyr2 + 4 | 2.1 ± 0.1 | 1.1 ± 0.1 |
| Tyr3 + 4 | 1.2 ± 0.1 | 2.4 ± 0.1 |
| Tyr4 + 5 | 1.4 ± 0.2 | 1.2 ± 0.1 |
| Tyr4 + 6 | 2.4 ± 0.2 | 1.2 ± 0.1 |

*Table 2 continued on next page*

*Table 2 continued*

| Substrate | $V_{max}$ (min$^{-1}$ hex$^{-1}$) | $K_M$ (µM) |
|-----------|-----------------------------------|------------|
| Tyr4 + 7 | 1.0 ± 0.1 | 0.67 ± 0.05 |
| Tyr4 + 8 | 2.0 ± 0.3 | 0.92 ± 0.06 |
| Tyr1 + 3 | 1.4 ± 0.1 | 0.9 ± 0.1 |
| Tyr2 + 3 | 0.81 ± 0.03 | 1.1 ± 0.1 |
| Tyr3 + 5 | 0.7 ± 0.1 | 0.43 ± 0.03 |
| Tyr3 + 6 | 1.1 ± 0.1 | 0.62 ± 0.05 |
| Tyr3 + 7 | 0.97 ± 0.08 | 0.61 ± 0.03 |
| Tyr3 + 8 | 0.49 ± 0.07 | 0.39 ± 0.06 |
| Val1 + 4 | 2.3 ± 0.1 | 1.6 ± 0.1 |
| **Val2 + 4** | 1.8 ± 0.1 | 1.2 ± 0.1 |
| **Val3 + 4** | 1.6 ± 0.1 | 1.1 ± 0.1 |
| Val4 + 5 | 2.1 ± 0.1 | 1.5 ± 0.1 |
| Val4 + 6 | 1.4 ± 0.1 | 1.1 ± 0.1 |
| Val4 + 7 | 1.3 ± 0.1 | 0.93 ± 0.01 |
| Val4 + 8 | 1.0 ± 0.1 | 0.88 ± 0.06 |

DOI: https://doi.org/10.7554/eLife.46808.005

substrates (*Figure 3E*) were inversely correlated with their energetic efficiencies of degradation (*Figure 3F*), indicating that poor grip results in non-productive ATP hydrolysis.

## Synergistic side-chain interactions promote GFP unfolding

Placing a single alanine at cassette position four with glycines at the remaining positions resulted in only marginally better degradation than the Gly$_{12}$ substrate (*Figure 3A and C*). By contrast, the GA cassette – with alanines at positions 1, 3, 7, 9, and 12 – supported efficient degradation (*Figure 1D and F*), despite the fact that positions 1, 7, 9, and 12 do not seem to be important determinants of grip (*Figure 4A*). This discrepancy suggested that synergistic interactions between the ClpX pore and multiple side chains might allow substantially better grip. To test this model, we constructed a panel of substrates with one alanine at position 4 and a second alanine at position 1, 2, 3, 5, 6, 7, 8, 9, 10, 11, or 12 (*Figure 4B*). In our cellular assay, alanines at cassette positions 1/4, 2/4, 3/4, and 4/5 supported robust degradation, alanines at positions 4/6 and 4/7 facilitated moderate degradation, and alanines at positions 4/8, 4/9, 4/10, 4/11 and 4/12 were little better than the single alanine at position 4 (*Figure 4B*). A single Ala at position 1 supported slightly better degradation in vivo than a single Ala at position 4, but the Ala-1/4 substrate was degraded more efficiently (*Figure 4C*). This difference was more pronounced in assays of degradation in vitro (*Figure 4D*, *Table 2*). Indeed, $V_{max}$ for degradation of the Ala-1/4 substrate (2.3 ± 0.2 min$^{-1}$) was more that 10-fold greater than $V_{max}$ for the Ala-1 substrate (0.19 ± 0.04 min$^{-1}$) or Ala-4 substrate (0.13 ± 0.06 min$^{-1}$). The non-additivity of these $V_{max}$ values provides direct evidence for synergy in grip.

Because the pore loops of ClpX and other AAA+ motors interact with every other substrate residue in cryo-EM structures (*Monroe et al., 2017*; *Gates et al., 2017*; *Puchades et al., 2017*; *de la Peña et al., 2018*; *Majumder et al., 2019*; *Dong et al., 2019*; *White et al., 2018*), we investigated whether a similar spacing of large side chains enhances grip. We designed panels of substrates with either Tyr or Val fixed at tail position 4 and a second residue of the same type at positions 1, 2, 3, 5, 6, 7, or 8 in an otherwise all-Gly cassette (*Figure 4E and F*). Among the Tyr substrates, the Tyr-2/4, Tyr-4/6, and Tyr-4/8 substrates promoted faster GFP degradation than the parental Tyr-4 substrate, whereas the other substrates exhibited similar or slower degradation (*Table 2*; *Figure 4E*). It is noteworthy that although the effect of multiple residues was irrelevant for the Ala-4/8 substrate (*Figure 4B*), the Tyr-4/8 substrate was gripped well, suggesting that spacing of Tyr in multiples of two may allow ClpX to grip substrate in a preferred conformation. Among the Val substrates, Val-1/4 and Val-4/5 facilitated slightly faster GFP degradation than the parental Val-4 substrate,

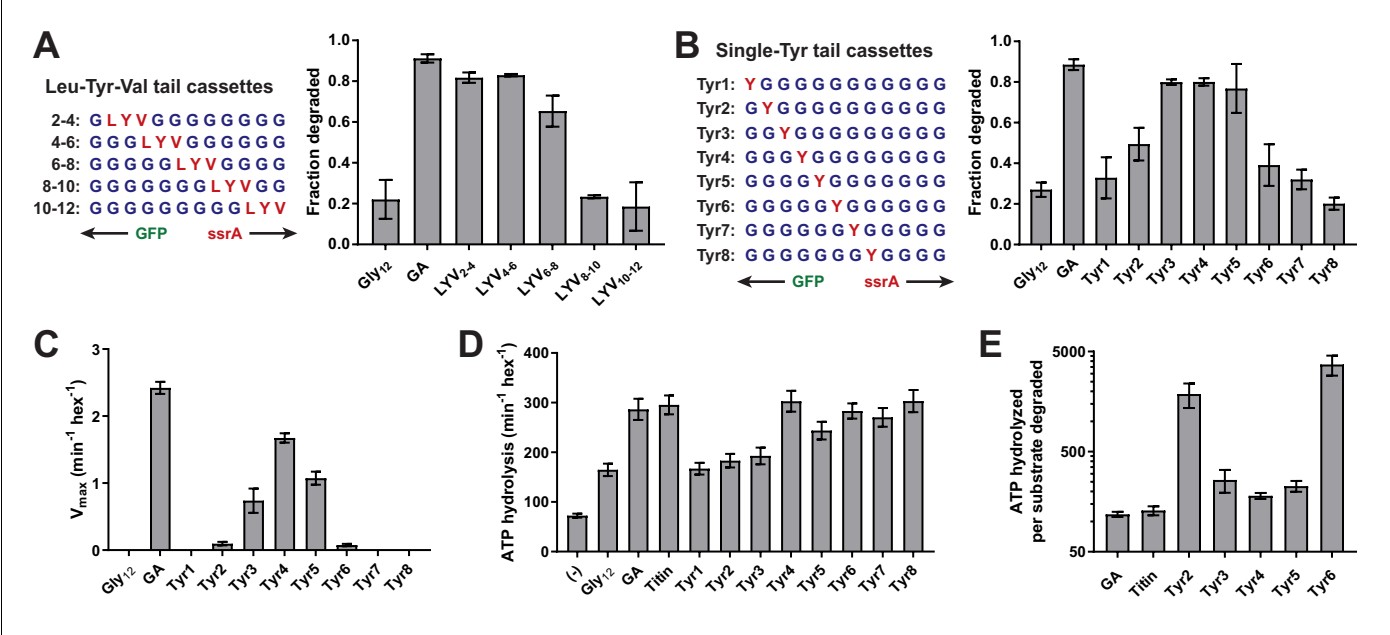

**Figure 2.** A small subset of tail residues mediate grip during GFP unfolding. (**A**) Fraction intracellular degradation for substrates with tails containing LYV tripeptides in otherwise all-glycine cassettes. $Gly_{12}$ and GA substrates were included as internal controls. (**B**) Fraction intracellular degradation for substrates with tails containing one tyrosine (Y) in otherwise all-glycine cassettes. $Gly_{12}$ and GA substrates were included as internal controls. (**C**) $V_{max}$ values from Michaelis-Menten analysis of degradation of purified substrates with single-tyrosine cassettes. (**D**) Rates of ATP hydrolysis by ClpX$^{\Delta N}$ (0.1 µM hexamer) in the presence of ClpP (0.3 µM 14-mer) in the absence (–) or presence of different substrates (15 µM monomer). (**E**) ATP cost of degrading substrates with single-tyrosine cassettes. Note that the Y-axis is logarithmic. In all panels, values represent averages (± S.D.) of three biological replicates.

DOI: https://doi.org/10.7554/eLife.46808.006

The following source data and figure supplements are available for figure 2:

**Source data 1.** Stimulation of ClpXP ATP hydrolysis by purified substrates.
DOI: https://doi.org/10.7554/eLife.46808.009

**Figure supplement 1.** Comparison of $K_M$ values for substrates tested in vitro; comparison of fitted values for $K_M$ for substrate degradation.
DOI: https://doi.org/10.7554/eLife.46808.007

**Figure supplement 2.** Stimulation of ClpXP ATP hydrolysis by purified substrates.
DOI: https://doi.org/10.7554/eLife.46808.008

Val-2/4 and Val-3/4 were degraded at similar rates to Val-4, and Val-4/6, Val-4/7, and Val-4/8 were degraded slightly slower (*Figure 4F*; *Table 2*). As the pattern of degradation rates for the branched Val residue (*Figure 4F*) is more similar to Ala (*Figure 4B and D*) than the aromatic Tyr (*Figure 4E*), it is possible that the ClpX pore-1 loops interact with aromatic residues somewhat differently from non-aromatic residues.

We tested an additional panel of substrates with a Tyr residue at position three and a second Tyr at positions 1, 2, 4, 5, 6, 7, or 8 (*Figure 4—figure supplement 1*). Unlike the other Tyr substrates, these substrates were generally degraded at rates slightly higher than those of the parental Tyr-3 substrate irrespective of spacing (*Table 2*; *Figure 4—figure supplement 1*). Thus, binding Tyr residues separated by multiples of two residues does not always enhance grip.

## Discussion

To unfold target proteins, ClpX and other AAA +protein remodeling machines use cycles of ATP binding and hydrolysis to pull on the degron tail of a substrate, thereby transmitting force to the native domain, but how these machines interact with individual tail residues during unfolding was poorly understood. Here, we identify and quantify the abilities of different tail residues to promote substrate grip during unfolding by ClpXP. Our experiments are enabled by the observation that placing 12 Gly residues between native GFP and a degron eliminates ClpXP degradation in vitro and

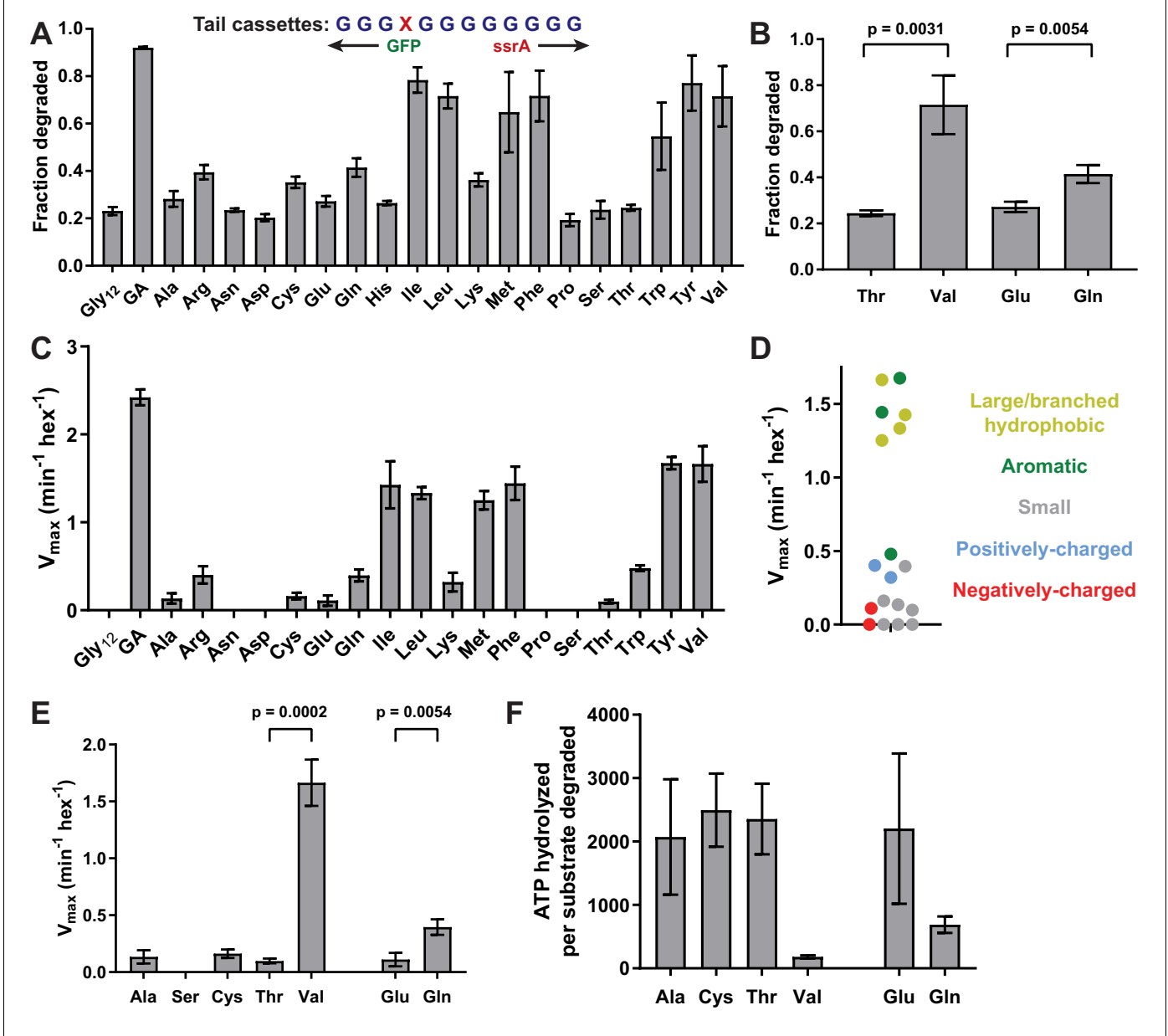

**Figure 3.** Side-chain grip effects at tail-position 4. (A) In substrates with otherwise all-glycine cassettes, fraction intracellular degradation depends on side-chain identity at tail-position 4. (B) Comparison of degradation in vivo for substrates with Thr or Val at tail-position four or Glu or Gln at tail-position 4 (Student's two-tailed t-test significance; Val/Thr: $t = 6.37$, df = 4; Glu/Gln: $t = 5.47$, df = 4). (C) $V_{max}$ values from Michaelis-Menten analysis of degradation of purified substrates. (D) Effects of position-4 residues, color-coded by side-chain properties, on $V_{max}$. (E) Comparison of degradation in vitro between substrates with Ala, Ser, Cys, Thr, or Val at tail-position four or Glu or Gln at tail-position 4 (Student's two-tailed t-test significance; Val/Thr: $t = 13.3$, df = 4; Glu/Gln: $t = 5.49$, df = 4). (F) ATP cost of degrading substrates with Ala, Cys, Thr, Val, Glu, or Gln at tail-position 4. With the exception of panel A, where $Gly_{12}$ and GA values represent averages (± S.D.) of nine biological replicates, all values represent three biological replicates.

DOI: https://doi.org/10.7554/eLife.46808.010

markedly slows degradation in vivo. GFP unfolding/degradation was not inhibited by 12-residue sequences containing mixtures of Gly and Ala (GA cassette); Gly, Lys, and Arg (basic cassette); or Gly, Asp, and Glu (acidic cassette). Compared with these sequences, ClpXP probably grips $Gly_{12}$ poorly because of the absence of β-carbons and distal side-chain atoms or increased backbone

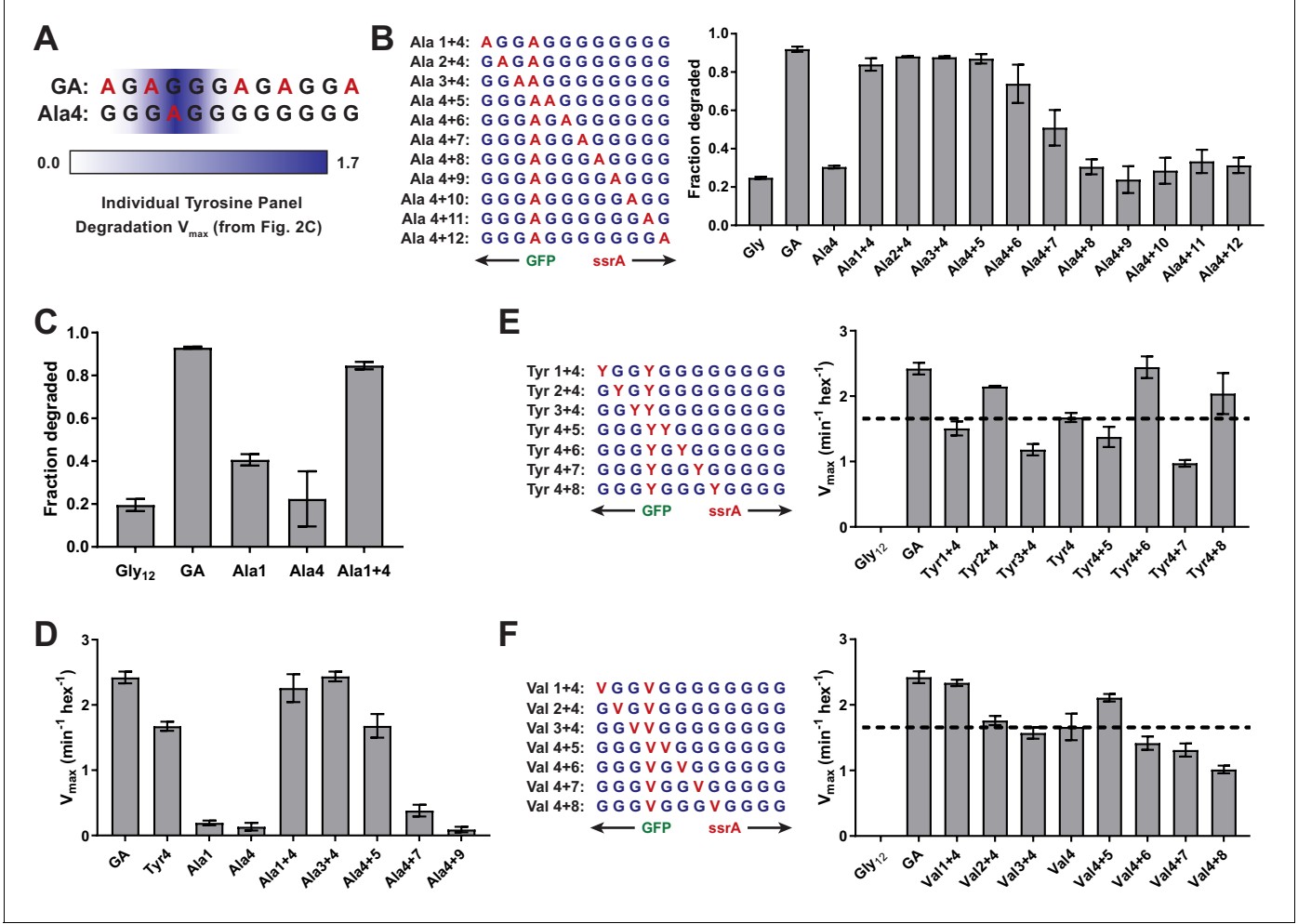

**Figure 4.** Multiple substrate residues contribute synergistically to grip. (**A**) GA and Ala-4 cassette sequences. A heatmap of $V_{max}$ values from *Figure 2C* is overlaid to show contribution of single tyrosine residues as each tail position. (**B**) Fraction intracellular degradation of substrates with one alanine at tail-position 4 and a second alanine at a variable position in otherwise all-glycine cassettes. (**C**) Comparison of intracellular degradation for a subset of substrates, including Ala-1. (**D**) $V_{max}$ values from Michaelis-Menten analysis of degradation of purified substrates. (**E and F**) Michaelis-Menten $V_{max}$ values for purified substrates with one tyrosine (**E**) or valine (**F**) at tail-position four and a second tyrosine (**E**) or valine (**F**) at each tail position in otherwise all-glycine cassettes. Overlaid dashed lines indicate degradation rate for the parental Tyr-4 (**E**) or Val-4 (**F**) substrates. In all panels, values represent averages (± S.D.) of three biological replicates.

DOI: https://doi.org/10.7554/eLife.46808.011

The following figure supplement is available for figure 4:

**Figure supplement 1.** Degradation of Dual-Tyr substrates centered at tail position 3.

DOI: https://doi.org/10.7554/eLife.46808.012

flexibility. We use 'grip' in a functional rather than strictly physical sense, although the two concepts are undoubtedly related.

Ensemble and single-molecule experiments show that ClpXP can translocate an enormous number of different amino-acid sequences, including long Gly tracts, with only minor velocity differences (*Aubin-Tam et al., 2011*; *Maillard et al., 2011*; *Sen et al., 2013*; *Cordova et al., 2014*; *Barkow et al., 2009*). For example, in assays requiring ATP-dependent translocation, ClpXP degraded peptide substrates containing $Gly_{10}$, $[Val-Gly]_5$, or $[Phe-Gly]_5$ sequences at similar rates (*Barkow et al., 2009*). However, if ClpX can translocate poly-Gly sequences, then why does $Gly_{12}$ inhibit or slow unfolding/degradation? When ClpX pulls on a native protein, Newtonian mechanics dictate that the folded domain resists with an opposing force, which would be absent during translocation of an unstructured polypeptide. Hence, when ATP hydrolysis is coupled to molecular motion

during an unfolding power stroke, we imagine that ClpX's grip on the $Gly_{12}$ sequence is insufficient to resist the opposition of the folded domain, causing an unproductive power stroke in which the pore-1 loops slip and fail to advance the substrate tail. In support of this model, we find that poor grip correlates with substantial increases in the ATP cost of degradation for the position-4 and Tyr-scan variants, an indication of slipping and futile power strokes. For example, degradation of one molecule of the Tyr-3, Tyr-4, and Tyr-5 substrates required hydrolysis of an average of ~180–240 ATPs, whereas degradation of the Tyr-2 or Tyr-6 substrates required hydrolysis of ~1900 and~3700 ATPs, respectively. These findings corroborate a previous report of futile power strokes during unsuccessful unfolding of a difficult substrate by ClpXP (*Kraut, 2013*). Furthermore, substrate-tail contacts with the axial pore that stimulate ATP hydrolysis by ClpX do not fully overlap with the contacts that determine grip.

A Tyr-scan experiment shows that tail-position 4 is most important for grip, with flanking positions showing diminishing effects. We expect that amino-acid substitutions at positions 3 and 5 would show side-chain grip trends similar to those observed at position 4. This is not true at tail-position 1, where Tyr did not improve grip but Ala did, perhaps because this part of the tail interacts with different residues in ClpX or the folded GFP domain than downstream positions. A single Ala at tail-position four is gripped poorly but a second Ala at certain positions can improve unfolding/degradation. Contacts between the second Ala and the ClpX pore may contribute to stronger grip. Alternatively, the second Ala might affect ClpX contacts made by the first Ala by altering the substrate conformation.

In our GFP substrate with a single Tyr at tail-position 4, this side chain is likely to contact a pore-1 loop close to the folded GFP domain. In an extended chain, four residues would span ~12 Å, a distance that modeling suggests would allow interaction with either the highest or second highest pore-1 loop of ClpX (*Figure 5A*; *Puchades et al., 2017*; X. Fei, personal communication). This is also an area where the axial channel is most tightly constricted around substrate. The distribution of Tyr-effects at positions 2–6 could reflect interactions with different pore-1 loops or the probability that Tyr side chains at different positions contact one specific pore-1 loop. Optical trapping studies indicate that 5–8 residues are moved by a single ClpXP power stroke (*Aubin-Tam et al., 2011*; *Maillard et al., 2011*). Thus, once the Tyr side chain at position 4 is engaged by a pore-1 loop, one successful translocation event probably unfolds GFP. Indeed, although many unsuccessful power

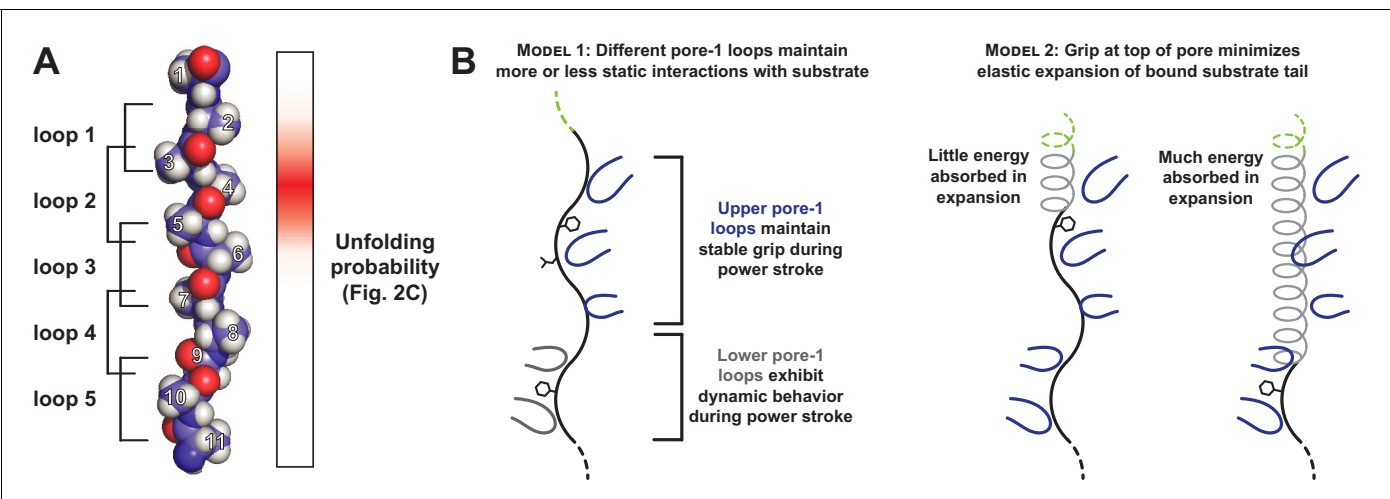

**Figure 5.** Only a subset of pore-1 loops in ClpX appear to mediate substrate grip. (**A**) Model of an extended poly-alanine substrate in the axial pore of ClpX and its interactions with different pore-1 loops based on cryo-EM structures of ClpXP (X.Fei, T.A. Bell, B.M. Stinson, S. Jenni, T.A. Baker, S.C. Harrison, and R.T. Sauer, in preparation). Similar loop-substrate interactions are observed in the yeast AAA+ protease Yme1 (*Puchades et al., 2017*). On the right, a heatmap of $V_{max}$ values from *Figure 2C* is shown. The substrate tail residues are numbered relative to where a folded domain would be expected to sit at the apical surface of the AAA+ ring during unfolding. Tail residues 2–6, which promote strong grip in ClpX, are positioned to interact with the three pore-1 loops at the top of the axial pore. (**B**) Two models for asymmetric contribution of pore-1 loops to substrate grip.
DOI: https://doi.org/10.7554/eLife.46808.013

strokes and ATP hydrolysis events occur while ClpXP is attempting to unfold a stable domain, hydrolysis of a single ATP ultimately results in unfolding. We find it notable that the Tyr-scan distribution is only two residues wide at half height. Hence, one tyrosine at position 4 mediates robust unfolding, whereas one tyrosine at position 7 has no effect. If a position-4 side chain can contact a pore-1 loop high in the ClpX pore, then a position-7 side chain should be able to contact another pore-1 loop lower in the pore. If this model is correct, then it implies that physical contacts between substrate tail residues and the upper pore-1 loops of ClpX are far more important for grip than interaction with the lower loops.

We can imagine several different mechanisms for the asymmetry in grip between pore-1 loops in the upper and lower sections of the ClpX pore. In one model, the stronger grip of upper pore-1 loops occurs because these loops maintain relatively static interactions with substrate throughout a power stroke (*Figure 5B*, left). Several translocation models have been recently proposed for AAA + unfoldases in which ATP-bound subunits with pore-1 loops oriented near the top of the pore move together as a rigid unit in response to ATP hydrolysis in a lower subunit (*Monroe et al., 2017*; *Puchades et al., 2017*; *de la Peña et al., 2018*; *Dong et al., 2019*). Furthermore, a previous study demonstrated that pore-1 loop mutations disrupt substrate unfolding most dramatically when they are in neighboring ClpX subunits, consistent with grip mediated by a clustered subset of pore-1 loops (*Iosefson et al., 2015*). Alternatively, the substrate tail in the pore could absorb some unfolding force through elastic expansion, diverting part of the energy of each power stroke away from unfolding (*Figure 5B*, right). Substrate interactions with the uppermost pore-1 loops would minimize the expansion length of the substrate tail, whereas interactions with lower pore-1 loops would allow the tail to absorb more force.

In an otherwise all-Gly cassette context, we find that Val, Ile, Leu, Met, Phe, and Tyr at tail-position 4 all promote reasonable levels of grip. If we think of poly-Gly as a smooth and relatively featureless rope, then these larger and generally non-polar side chains can be viewed as knots in the rope that afford better grip. However, grip is not a simple function of side-chain size. For example, Trp supports slower GFP unfolding than the better-gripped residues, suggesting that there may be an upper limit on the size of a side chain that can be efficiently gripped. Polar atoms, especially those close to the peptide backbone of the substrate, weaken grip. For example, Val is one of the best residues in terms of grip, whereas Thr, which differs only by substituting a hydroxyl for a methyl group, barely supports unfolding. Similarly, Ser alone does not support GFP unfolding, but removing the hydroxyl group (Ala) or substituting a less-polar thiol group (Cys) restores low-level unfolding activity. ClpXP may grip polar side chains less tightly because oxygen or nitrogen atoms bearing partial or full charges are not fully solvated when they are in productive contact with a pore-1 loop and thus incur an energetic penalty. Our finding that large hydrophobic and aromatic side chains are gripped well by ClpX is consistent with a model in which van der Waal's or hydrophobic interactions between the pore-1 loops and specific side chains in the tail are largely responsible for grip.

Several recent cryo-EM structures of AAA+ proteases and protein-remodeling motors reveal a spiral arrangement of subunits in which aromatic residues in the pore-1 loops interact with substrate side chains spaced two residues apart (*Monroe et al., 2017*; *Gates et al., 2017*; *Puchades et al., 2017*; *de la Peña et al., 2018*; *Majumder et al., 2019*; *Dong et al., 2019*; *White et al., 2018*). Our observation that substrate tails with two tyrosines can in some cases specifically enhance grip when spaced by a multiple of two residues is consistent with these structures. However, substrates with two Val residues do not exhibit the same periodic grip enhancement as Tyr, and multiple Ala residues together promote strong grip regardless of their relative spacing. It is clear that nonspecific interactions between the axial pore and substrate side chains are sufficient to promote strong grip independent of precise pore-1 loop intercalation. In specific cases, periodic side chain intercalation could enhance grip for aromatic side chains through the establishment of π-stacking networks with the pore-1 loop Tyr residues, possibly by optimizing the bound substrate conformation.

A recent cryo-EM structure of the AAA+ motor NSF, which disassembles SNARE complexes following vesicle fusion, contains well-resolved density for substrate side chains, revealing interactions with the pore-1 loop tyrosine (*White et al., 2018*). Although NSF disassembles SNARE complexes in a single round of ATP turnover in a mechanism distinct from ClpX (*Ryu et al., 2015*), the structural similarities between NSF and many AAA+ unfoldase proteases suggests a common mode of substrate interaction and grip. The assembled SNARE complex is remarkably stable, and NSF likely requires strong grip to disassemble the complex. Consistent with our biochemical observations for

ClpX, this structure indicates that the strongest substrate contacts are formed with pore-1 loops high in the upper ring of NSF, and that two substrate residues (Met and Leu) that are gripped well by ClpX participate in these interactions.

A previous study found that a $Gly_{15}$ sequence placed between GFP and an ssrA tag did not slow ClpXP degradation (*Barkow et al., 2009*). However, their substrate contained six additional residues ($Thr^1$-$His^2$-$Gly^3$-$Met^4$-$Asp^5$-$Glu^6$) between folded GFP and the $Gly_{15}$ sequence. As we find that Met at position four supports robust unfolding, it is likely that interactions with the extra sequence mediate unfolding of the $Gly_{15}$ substrate. Our findings suggest that evolutionary placement of tail residues that are gripped well by ClpX may tune degradation of substrates that unfold non-cooperatively or that have multiple-folded domains. Complete, processive unfolding of multi-domain substrates depends critically on interactions between ClpX and the peptide-tail remnants from unfolding and degradation of the previous domain. For example, ClpXP degradation of Domain III of *C. crescentus* DnaX is inhibited by a Gly-rich sequence between Domains III and IV, which acts as a partial processing mechanism essential for DNA replication (*Vass and Chien, 2013*). Gly-rich tracts also inhibit unfolding/degradation of *E. coli* DHFR, although multiple alanines in the tail do not improve degradation of this substrate (*Too et al., 2013*). Thus, ClpX unfolding of different native substrates probably requires different degrees of grip strength, which could be mediated by more and/or better-gripped amino acids adjacent to the folded domain.

ClpXP contains just two types of subunits, whereas the 26S proteasome consists of more than 30 subunit types (*Budenholzer et al., 2017*). Nevertheless, our work is reminiscent of and reinforces studies of proteasomal degradation by Matouschek and colleagues. For example, they find that low-complexity sequences primarily composed of Gly, Ser, or Thr residues can inhibit proteasomal degradation (*Tian et al., 2005*); these residues individually are also insufficient to promote ClpXP degradation of GFP. Similarly, sequences that include Phe and Tyr residues can improve or rescue degradation by both the proteasome and ClpXP. These similarities may arise because the $Rpt_{1-6}$ unfolding ring of the proteasome, despite containing six distinct subunits, has pore-1 loops very similar to those of ClpX. Given the structural similarities between many AAA+ protein remodeling machines, we expect that the principles underlying grip in ClpX reflect those of the broader family.

## Materials and methods

### Key resources table

| Reagent type (species) or resource | Designation | Source or reference | Identifiers | Additional information |
|---|---|---|---|---|
| Cell line (*Escherichia coli*) | *E. coli* T7 Express ΔclpA ΔclpP ΔclpX | this paper | | *E. coli* strain lacking the ClpA, ClpP, and ClpX genes. progenitor: *E. coli* T7 Express (New England Biolabs #C2566) |
| Recombinant DNA reagent | pT7 ClpX$^{ΔN}$ (plasmid) | *Martin et al., 2005* | | N-terminally $His_6$-tagged ClpX$^{ΔN}$ (residues 62–424) for overexpression |
| Recombinant DNA reagent | pT7 ClpP (plasmid) | *Kim et al., 2000* | | C-terminally $His_6$-tagged ClpP for overexpression |
| Recombinant DNA reagent | pBAD ClpP/ClpX$^{ΔN}$ (plasmid) | this paper | | for inducible polycistronic expression of ClpP and ClpX$^{ΔN}$ (residues 62–424) for cytoplasmic GFP degradation assays. Progenitor: pBAD (*Guzman et al., 1995*; jb.177.14.4121–4130.199) |

*Continued on next page*

Continued

| Reagent type (species) or resource | Designation | Source or reference | Identifiers | Additional information |
|---|---|---|---|---|
| Recombinant DNA reagent | pBAD null (plasmid) | this paper | | control plasmid for cytoplasmic GFP degradation assays. Progenitor: pBAD (*Guzman et al., 1995*) |
| Recombinant DNA reagent | ProD GFP Gly12 ssrA (plasmid) | this paper | | for constitutive expression of GFP (residues 1–229) substrates with a 12xGly cassette and partial ssrA (GSENYALAA). All other substrates are derivatives of this construct with different variable cassette sequences. Progenitor: ProD Gemini (*Davis et al., 2011*; nar/gkq81) |
| Recombinant DNA reagent | pT7 GFP Gly12 ssrA (plasmid) | this paper | | for overexpression of N-terminally His$_6$-tagged GFP (1-229) substrates with a 12xGly cassette and partial ssrA (GSENYALAA). All other substrates are derivatives of this construct with different variable cassette sequences. |

## Plasmid and strain construction

An expression plasmid containing *E. coli* ClpP and *E. coli* ClpX$^{\Delta N}$ was constructed by cloning ClpP into the open reading frame downstream of the pBAD promoter in pBAD18 (*Guzman et al., 1995*). A second ribosome binding site (5'-CAAGGAGAATAACG-3') and the ClpX$^{\Delta N}$ coding sequence (residues 62–424) was added downstream of the ClpP stop codon to produce a polycistronic expression construct. GFP substrates for cytoplasmic degradation assays were cloned downstream of the constitutive insulated ProD promoter in pSB3C5 (*Davis et al., 2011*). His$_6$-GFP substrates for purification were cloned into a pET4b derivative downstream of the pT7 promoter. For all substrates, the 12-residue variable cassette was encoded on an oligonucleotide and introduced upstream of a partial ssrA degron (Gly-Ser-Glu-Asn-Tyr-Ala-Leu-Ala-Ala) using PCR mutagenesis. The seven C-terminal residues of this degron are identical to those of the ssrA tag, but we removed the N-terminal part of the ssrA tag to preclude potential SspB inhibition (*Hersch et al., 2004*).

T7 Express Δ*clpA* Δ*clpP* Δ*clpX* was generated from the *E. coli* strain T7 Express (New England Biolabs). The bicistronic *clpP*–*clpX* locus was removed by using lambda *red* recombineering (*Yu et al., 2000*) to replace the locus with an FRT-Kan$^R$ cassette, which was subsequently removed by FLP recombinase expression. ClpA::FRT-Kan$^R$ was then transduced into this strain with P1 phage from a ClpA::FRT-Kan$^R$ strain in the Keio collection (*Baba et al., 2006*), and the resistance marker was again removed with FLP recombinase. Modification of the correct loci was verified by PCR at each step in strain construction.

## Protein expression and purification

His$_6$-GFP-cassette-ssrA constructs were expressed as described (*Kim et al., 2000*) and purified by Ni-NTA affinity, Source 15Q anion exchange, and Superdex 200 size-exclusion chromatography. Purified substrates were assessed to be >99% pure by SDS-PAGE and were stored in 25 mM HEPES-KOH (pH 7.5), 150 mM KCl, 10% glycerol, and 500 μM dithiothreitol.

## Degradation assays in vivo

The ProD-GFP plasmid encoding each substrate was transformed into T7 Express $\Delta clpA$ $\Delta clpP$ $\Delta clpX$ cells carrying either pBAD18(ClpP/ClpX$^{\Delta N}$) or pBAD18(null). After overnight growth at 30°C on LB agar plates supplemented with 100 µg/mL ampicillin and 34 µg/mL chloramphenicol, single colonies were picked into 5 mL of the same medium and antibiotics and cultures were grown overnight at 30°C. At the start of degradation assays, 50 µL of culture of either the ClpP/ClpX$^{\Delta N}$ expression strain or the null control strain for each substrate was inoculated into fresh 5 mL LB plus antibiotics and grown at 37°C to OD$_{600}$ 0.7–1.0. The cultures were then centrifuged; resuspended at OD$_{600}$ 1.2 in fresh media plus antibiotics; and 500 µL was added to 1 mL of fresh media plus antibiotics supplemented with 120 mM L-arabinose, for a final concentration of 80 mM L-arabinose and OD$_{600}$ of 0.4. After 35 min of growth at 37°C, 1 mL of culture was removed, centrifuged, and resuspended in 600 µL of phosphate buffered saline (pH 7.4). Three 150 µL technical replicates of resuspended cells were transferred to wells of a clear-bottom black 96-well plate (Greiner). Both the GFP fluorescence of the cell resuspension (excitation 467 nm, emission 511 nm) and the optical density (absorbance 600 nm) were measured on a SpectraMax M5 plate reader (Molecular Devices). The GFP fluorescence for each ClpX$^{\Delta N}$P sample and control sample was divided by the measured cell density to give normalized fl$_{protease}$ and fl$_{control}$ values respectively. Fraction degraded was calculated as:

$$1 - (\mathrm{fl_{protease}}/\mathrm{fl_{control}})$$

Each degradation assay was performed independently in multiple biological replicates, and the calculated value of fraction degraded was averaged across biological replicates. No obvious outliers were observed, and all values were included in the subsequent analysis.

## Biochemical assays in vitro

Degradation assays were performed at 37°C in 25 mM HEPES-KOH (pH 7.5), 5 mM MgCl$_2$, 200 mM KCl, 10% glycerol, with 0.1 µM ClpX$^{\Delta N}$ (hexamer), 0.3 µM ClpP (14-mer), 5 mM ATP, 32 mM creatine phosphate (Roche), and 0.08 mg/mL creatine kinase (Millipore-Sigma). For the Thr-4 substrate, assays were performed with 0.5 µM ClpX$^{\Delta N}$ (hexamer) and 1.5 µM ClpP (14-mer) to measure degradation rates more accurately and facilitate comparison with substrates that were degraded more rapidly. Degradation rates were measured by decrease in fluorescence (excitation 467 nm; emission 511 nm) in 20 µL reactions on a SpectraMax M5 plate reader. To control for signal loss from photobleaching, a parallel set of reactions was measured for each substrate without ClpX$^{\Delta N}$ or ClpP, and changes in GFP fluorescence in this experiment were subtracted from those in the degradation reaction. Each measurement included three technical replicates measured together in parallel, and the average values of these replicates were fit to a hyperbolic equation to determine $K_M$ and V$_{max}$. Three independently-conducted biological replicates were performed in this manner for each substrate to determine average values (± S.D.) for $K_M$ and V$_{max}$. No obvious outliers were observed, and all values were included in the subsequent analysis.

ATP hydrolysis rates were measured using a coupled-NADH oxidation assay as described (*Martin et al., 2005*). Degradation efficiency (ATP hydrolyzed per substrate degraded) was calculated by dividing the rate of ATP hydrolysis at a near saturating substrate concentration (15 µM) by V$_{max}$ for substrate degradation.

## Acknowledgements

This work was supported by US National Institutes of Health (NIH) grant GM-101988 (RTS). TA Bell was supported in part by US NIH grant 5T32GM-007287. TA Baker is an employee of the Howard Hughes Medical Institute. We thank X Fei (MIT) for providing information on the ClpXP cryo-EM structure and present and past members of the Sauer and Baker labs for helpful conversations and feedback on the manuscript.

## Additional information

### Funding

| Funder | Grant reference number | Author |
| --- | --- | --- |
| National Institutes of Health | GM-101988 | Robert T Sauer |
| National Institutes of Health | 5T32GM-007287 | Tristan A Bell |
| Howard Hughes Medical Institute | | Tania A Baker |

The funders had no role in study design, data collection and interpretation, or the decision to submit the work for publication.

### Author contributions
Tristan A Bell, Conceptualization, Data curation, Formal analysis, Validation, Investigation, Visualization, Methodology, Writing—original draft, Writing—review and editing; Tania A Baker, Supervision, Funding acquisition; Robert T Sauer, Supervision, Funding acquisition, Project administration, Writing—review and editing

### Author ORCIDs
Tristan A Bell (iD) https://orcid.org/0000-0002-3668-8412
Tania A Baker (iD) http://orcid.org/0000-0002-0737-3411

### Decision letter and Author response
Decision letter https://doi.org/10.7554/eLife.46808.016
Author response https://doi.org/10.7554/eLife.46808.017

## Additional files

### Supplementary files
• Transparent reporting form
DOI: https://doi.org/10.7554/eLife.46808.014

### Data availability
All data generated during this study are included in the manuscript and supporting files as Tables 1 and 2 and Figure 2 - source data 1.

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
