## [Decision Letter]

Thank you for submitting your article "Interactions between a subset of substrate side chains and AAA+ motor pore loops determine grip during protein unfolding" for consideration by *eLife*. Your article has been reviewed by three peer reviewers, one of whom is a member of our Board of Reviewing Editors, and the evaluation has been overseen by John Kuriyan as the Senior Editor. The following individual involved in review of your submission has agreed to reveal his identity: Gabriel C Lander (Reviewer #1).

The reviewers have discussed the reviews with one another and the Reviewing Editor has drafted this decision to help you prepare a revised submission.

Summary:

In the manuscript by Bell et al., the authors implement a biochemical system to characterize how different types of residues at 12 positions preceding a folded domain (GFP) influence substrate unfolding by the AAA+ protease ClpXP. This study demonstrates that different types of amino acids substantially impact unfolding, with generally consistent results in vitro and in vivo. The results show that the grip produced by the conserved pore residues of the ClpX AAA ATPase depends on intercalating sidechains, i.e., a poly-gly chain does not produce any grip. A single large side chain within a series of Gly residues is often sufficient, with some notable exceptions, e.g., charged or polar residues within poly-Gly are not as efficient.

This work is particularly relevant in the context of a substrate ATPase interaction that appears to be conserved across the AAA+ superfamily based on several cryo-EM structures. However, the molecular and enzymatic implications of this mechanism for substrate processing are still unclear.

Essential revisions:

1) The authors conclude that substrate grip is largely dictated by van der Waals interactions, since bulkier residues are generally associated with a stronger grip on substrate. The authors also mention that polar side chains may result in a weaker grip due to partial solvation. However, it remains unclear why valine residues within a substrate would produce stronger effects on unfolding than the significantly bulkier aromatic residues Phe or Trp. Further, the observation that Ile and Leu produce grips that are similar to Phe and Tyr is puzzling due to the different sizes of these residues. Please comment and discuss.

2) The rationale for using a Leu-Tyr-Val tail cassette (Figure 2A) is unclear since this tripeptide is found in ClpX loop-1 and not in substrates..

3) Is it known that happens to the ssrA degron? Is it "pulled" through ClpX? Is the presence of the ssrA degron absolutely required for the assay in Figure 1, and, if so, where does this specificity come from? Is this perhaps needed for a search process preceding the loaded, "gripped" state. It would be interesting (but optional) to explore the effects of glycines and tyrosines in the ssrA tail using this system. Please comment and discuss.

4) It is unclear whether an unfolded Xxx12 tail preceding the degron motif is found in substrates in the cell. Please comment and discuss.

5) How might cassette charge affect the steps preceding gripping that would explain why single charged residues within Gly residues do not provide good grip? Could this effect be related a process involving non-specific, pre-loading interactions with ClpX, either in the basket-like region around the pore or the split region if there is some sort of side-loading happening?

6) Discussion: "However, if ClpX can translocate poly-Gly sequence… Hence, during an unfolding power stroke, we imagine that the resisting force weakens ClpX's grip on the Gly12 sequence, resulting in ineffective power strokes that fail to move the substrate tail.". A possible explanation is that the ClpX pore loops only engage 11 residues (Figure 5 according to the unpublished CryoEM structure). Thus, it requires at least one non-Gly residue in a 12 residue stretch, so the ClpX motor can pull on the substrate, otherwise, there is "slippage".

7) Figure 1C-why is the fluorescence signal so low in the -ClpX(deltaN)P case for titin (by nearly an order of magnitude lower than the others)?

8) Panels 2C, D, E-please provide Gly12/GA controls here as well.

9) Panel 2E. Please provide the number of hydrolyzed ATPs for the titin as well. The requirement of thousands of ATP is interesting. In contrast, NSF can disassemble the SNARE complex with just 12 ATPs (Ryu et al., 2015).

10) Figure 3C. Please provide Gly12/GA controls, and also test serine.

11) Figure 3F. The error bar for Thr is very large. Why?

12) White et al., 2018, showed that the SNARE substrate is engaged in NSF via very similar interactions as suggested in Figure 5 (according to the unpublished CryoEM structure of CplX-substrate), and similar to Yme1. Interestingly, White et al., observed that SNARE substrate is loaded into the pore in the ATP state of NSF without hydrolysis, although the loading requires some unfolding of the substrate for this loading process. Thus, the "grip" mechanism may also be important for an initial pre-hydrolysis loading step. Moreover, the two of the most well-defined interactions with the SNARE SNAP-25 N terminus are a methionine and leucine (consistent with the findings in this work), although there are charged residues interspersed as well. Please comment and discuss.

13) Figure 5. Please note that NSF/SNAP/SNARE CryoEM structure by White et al., 2018, has some of the best resolved substrate sidechain densities for the substrate among published AAA-substrate complexes.

Optional:

14) Within the spiral staircase of ATPases, the pore-loops of multiple subunits engage substrate synergistically, and the authors establish a synergistic effect in the presence of multiple alanine residues. An optional experiment that tests the synergistic effect of multiple tyrosine residues would be an impactful addition to the manuscript since, given the organization of the substrate within the staircase, it is reasonable to believe that alternating aromatic residues in the substrate might engage in pi-stacking interactions with the pore loop aromatics.

---

## [Author Response]

Essential revisions:1) The authors conclude that substrate grip is largely dictated by van der Waals interactions, since bulkier residues are generally associated with a stronger grip on substrate. The authors also mention that polar side chains may result in a weaker grip due to partial solvation. However, it remains unclear why valine residues within a substrate would produce stronger effects on unfolding than the significantly bulkier aromatic residues Phe or Trp. Further, the observation that Ile and Leu produce grips that are similar to Phe and Tyr is puzzling due to the different sizes of these residues. Please comment and discuss.

Our results show that Val and most larger hydrophobic/aromatic residues at position 4 promote reasonable grip, which we now make clearer in the text (Discussion paragraph seven). The differences between Val, Ile, Leu, Met, Phe, and Tyr (Figure 3C) are relatively minor and could be due to many factors, including rotamer preferences, packing efficiency, effects on secondary-structure propensity, desolvation effects, etc. In the absence of high-resolution structures of appropriate complexes, we do not feel that speculating on these factors is warranted. Trp is substantially poorer than Phe/Tyr, which we now note in the Discussion and state that this suggests an upper limit on side-chain size for optimal gripping. We also expanded our substrate comparison in Figures 3E and 3F to include Ala, Ser, and Cys in addition to Thr and Val. The new results strengthen our observation that polar atoms or groups have unfavorable effects on grip for smaller side chains and are addressed in the revised Results subsection “Side-chain grip preferences” and Discussion paragraph seven.

2) The rationale for using a Leu-Tyr-Val tail cassette (Figure 2A) is unclear since this tripeptide is found in ClpX loop-1 and not in substrates..

We initially selected Leu-Tyr-Val because its large and hydrophobic residues were chemically very distinct from Gly-Gly-Gly. Serendipitously, this motif worked well. The similarity with the GYVG pore-1 loops in ClpX is coincidental. We’ve added text to the Results to clarify the rationale of our choice of tripeptide motif (see subsection “A small stretch of tail residues mediates unfolding grip”). We have also changed “YVG loops” to “pore-1 loops” in the text to avoid any confusion.

3) Is it known that happens to the ssrA degron? Is it "pulled" through ClpX? Is the presence of the ssrA degron absolutely required for the assay in Figure 1, and, if so, where does this specificity come from? Is this perhaps needed for a search process preceding the loaded, "gripped" state. It would be interesting (but optional) to explore the effects of glycines and tyrosines in the ssrA tail using this system. Please comment and discuss.

The mechanism of ssrA recognition and processing has been characterized previously (Flynn et al., PNAS, 2001; Martin et al., 2008). The C-terminal part of the ssrA sequence is required for recognition, is the first sequence to transit the pore and be degraded by ClpP, and is critical for the experiments presented. Although not reported in our paper, we confirmed early in the design of this project that GFP with a mutant C-terminal ssrA tag (AANDENYALAA mutated to AANDENYALDD) was not degraded by ClpXP, consistent with what has been reported elsewhere (Flynn et al., PNAS, 2001). As this paper focuses on grip during unfolding, we feel that an interrogation of ssrA specificity would be beyond the scope of this work. We have, however, explicitly noted the role of the ssrA tag in the Results to improve clarity for readers (see subsection “Substrate design and degradation assays”).

4) It is unclear whether an unfolded Xxx12 tail preceding the degron motif is found in substrates in the cell. Please comment and discuss.

For a single-domain protein, it is unlikely that an X_12_-degron tail would be present for ClpXP to interact with during unfolding. In multidomain substrates, however, the unfolding of all domains after the first will involve ClpXP interaction with a peptide tail formed from the remnants of the previous unfolded domain (Kenniston et al., PNAS, 2005) and poor grip has been implicated in partial processing of multidomain substrates (Kraut, 2013; Vass and Chien, 2013). We have included additional text in the Discussion to clarify this point for readers (paragraph ten).

5) How might cassette charge affect the steps preceding gripping that would explain why single charged residues within Gly residues do not provide good grip? Could this effect be related a process involving non-specific, pre-loading interactions with ClpX, either in the basket-like region around the pore or the split region if there is some sort of side-loading happening?

Our substrates with six acidic or six basic residues in the variable cassette were robustly unfolded and degraded by ClpXP (Figures 1D, 1F), suggesting that charge *per se* is not an issue. As noted above, binding of our substrates to ClpXP is mediated by the partial ssrA degron and not by the X_12_ cassette. ClpX functions as a topologically closed hexamer (Glynn et al., NSMB, 2012; Bell et al., Biochemistry, 2018) and side-loading is therefore unlikely.

6) Discussion: "However, if ClpX can translocate poly-Gly sequence… Hence, during an unfolding power stroke, we imagine that the resisting force weakens ClpX's grip on the Gly12 sequence, resulting in ineffective power strokes that fail to move the substrate tail.". A possible explanation is that the ClpX pore loops only engage 11 residues (Figure 5 according to the unpublished CryoEM structure). Thus, it requires at least one non-Gly residue in a 12 residue stretch, so the ClpX motor can pull on the substrate, otherwise, there is "slippage".

We agree that the phrasing was ambiguous and have revised the text to improve clarity (Discussion paragraph two).

7) Figure 1C-why is the fluorescence signal so low in the -ClpX(deltaN)P case for titin (by nearly an order of magnitude lower than the others)?

Expression of GFP substrates in *E. coli* was sensitive to the C-terminal cassette sequence as shown in Figure 1C as well as for yields of overexpressed GFP substrates (10-20 fold lower yield from overexpression than substrates with Gly-rich tails). This reduced expression might be caused by changes in mRNA stability or altered rates of protein synthesis. Because we observe rapid degradation of purified substrate in vitro and endogenously expressed substrate in vivo, we do not believe that the lower expression of this substrate impairs our measurement of grip during unfolding. To improve clarity for readers, we have expanded on the unexplained observation of lower expression for this substrate in the Results (subsection “Substrate design and degradation assays”).

8) Panels 2C, D, E-please provide Gly12/GA controls here as well.

We have now included these controls as requested. For Figure 2E, Gly_12_ was not included because efficiency could not be calculated for substrates that were not measurably degraded.

9) Panel 2E. Please provide the number of hydrolyzed ATPs for the titin as well. The requirement of thousands of ATP is interesting. In contrast, NSF can disassemble the SNARE complex with just 12 ATPs (Ryu, et al., 2015).

As requested, we have included these results in the revised Figures 2D and 2E. The titin-based tail is degraded with similar efficiency to the GA substrate, with a bulk energy use of ~150 ATP per substrate.

10) Figure 3C. Please provide Gly12/GA controls, and also test serine.

As requested, we have now included the Gly_12_/GA controls here, and also included Ser, Cys, and Pro substrates.

11) Figure 3F. The error bar for Thr is very large. Why?

The GFP-degradation assay has an across-the-board minimum error of ~0.03 substrate degraded min^-1^ due to measurement noise, pipetting variability, etc. Because efficiency is calculated as ATP hydrolysis per substrate degraded, the large relative error in measuring degradation V_max_ for the Thr-4 substrate (0.04 ± 0.03 min^-1^) propagated to the efficiency value. To determine more accurate rates, we re-measured degradation of the Thr-4 substrate using a 5-fold higher concentration of ClpXP, which substantially improved the Michaelis-Menten fits and errors. We have included a note in the Materials and methods that Thr-4 was measured at higher enzyme concentration than other substrates.

12) White et al., 2018, showed that the SNARE substrate is engaged in NSF via very similar interactions as suggested in Figure 5 (according to the unpublished CryoEM structure of CplX-substrate), and similar to Yme1. Interestingly, White et al., observed that SNARE substrate is loaded into the pore in the ATP state of NSF without hydrolysis, although the loading requires some unfolding of the substrate for this loading process. Thus, the "grip" mechanism may also be important for an initial pre-hydrolysis loading step. Moreover, the two of the most well-defined interactions with the SNARE SNAP-25 N terminus are a methionine and leucine (consistent with the findings in this work), although there are charged residues interspersed as well. Please comment and discuss.

Thank you for drawing our attention to this work. We have now referenced this NSF structure in the Introduction, and included a comparison of ClpX and NSF in the Discussion.

13) Figure 5. Please note that NSF/SNAP/SNARE CryoEM structure by White et al., 2018, has some of the best resolved substrate sidechain densities for the substrate among published AAA-substrate complexes.

We now include this fact in the Discussion.

14) Within the spiral staircase of ATPases, the pore-loops of multiple subunits engage substrate synergistically, and the authors establish a synergistic effect in the presence of multiple alanine residues. An optional experiment that tests the synergistic effect of multiple tyrosine residues would be an impactful addition to the manuscript since, given the organization of the substrate within the staircase, it is reasonable to believe that alternating aromatic residues in the substrate might engage in pi-stacking interactions with the pore loop aromatics.

We thank the reviewers for this idea. The revised manuscript contains data for several new substrates in which we explore different spacings of two tyrosines or two valines on degradation in vitro (new Figures 4E, 4F; Figure 4–figure supplement 1). The Tyr data hints at substrate side-chain intercalation with pore loops enhancing grip, possibly through π-stacking with pore-loop Tyr residues, which we address in the revised Results and Discussion.

References:

Aubin-Tam, M.E., Olivares, A.O., Sauer, R.T., Baker, T.A., and Lang, M.J. (2011). Single-molecule protein unfolding and translocation by an ATP-fueled proteolytic machine. Cell *145,* 257–267.

Bell, T.A., Baker, T.A., and Sauer, R.T. (2018). Hinge-Linker Elements in the AAA+ protein unfoldase ClpX mediate intersubunit communication, assembly, and mechanical activity. Biochemistry *57*, 6787-6796.

Flynn, J.M., Levchenko, I., Seidel, M., Wickner, S.H., Sauer, R.T., and Baker, T.A. (2001). Overlapping recognition determinants within the ssrA degradation tag allow modulation of proteolysis. Proc Nat Acad Sci *98*, 10584-10589.

Glynn, S.E., Nager, A.R., Baker, T.A., and Sauer, R.T. (2012). Dynamic and static components power unfolding in topologically closed rings of a AAA+ proteolytic machine. Nat Struct Mol Biol *19*, 616-622.

Kenniston, J.A., Baker, T.A., and Sauer, R.T. (2005). Partitioning between unfolding and release of native domains during ClpXP degradation determines substrate selectivity and partial processing. Proc Nat Acad Sci *102*, 1390-1395.

Kraut, D.A. (2013). Slippery substrates impair ATP-dependent protease function by slowing unfolding. J Biol Chem *288,* 34729–34735.

Martin, A., Baker, T.A., and Sauer, R.T. (2008). Diverse pore loops of the AAA+ ClpX machine mediate unassisted and adaptor-dependent recognition of ssrA-tagged substrates. Mol Cell *29,* 441–450.

Sen, M., Maillard, R.A., Nyquist, K., Rodriguez-Aliaga, P., Pressé, S., Martin, A., and Bustamante, C. (2013). The ClpXP protease unfolds substrates using a constant rate of pulling but different gears. Cell *155,* 636–646.

Vass, R.H., and Chien, P. (2013). Critical clamp loader processing by an essential AAA+ protease in *Caulobacter crescentus*. Proc Nat Acad Sci *110,* 18138–18143.